# A Neurosymbolic Agent System for Compositional Visual Reasoning

## Abstract

The advancement in large language models (LLMs) and large vision models has fueled the rapid progress in multi-modal vision-language reasoning capabilities. However, existing vision–language models (VLMs) remain challenged by compositional visual reasoning. This paper presents VLAgent, a neuro-symbolic approach to developing a Vision-Language Agent system for efficient compositional visual reasoning with three novel features. *First*, VLAgent develops an interpretable visualization-enhanced two-stage neuro-symbolic reasoning system. The first stage is managed by a front-end engine that generates a structured visual reasoning plan (symbolic program script) for each compositional visual reasoning task by utilizing a pre-trained LLM powered with few-shot chain-of-thought in-context learning. The second stage is managed by a high-performance back-end engine. It transforms the planning script into executable code based on visual input (image or video) and the combination of neural models and symbolic functions and then performs a sequence of actions for the compositional visual reason task. *Second*, to ensure and enhance the quality of mapping the logic plan to a sequence of executable instructions, VLAgent introduces the SS-parser, which examines the syntax and semantic correctness of the planning script, detects and repairs the logic errors found in the LLM-generated logic plan before generating the executable program. *Third*, VLAgent introduces the execution verifier in critical reasoning steps to validate and refine its compositional reasoning results in a stepwise manner, for example, ensemble methods for critical visual reasoning and caption analysis for low-confidence compositional reasoning. Extensive experiments were conducted on six visual benchmarks and compared to a dozen SoTA visual reasoning models. The results show that VLAgent outperforms existing representative approaches to compositional visual reasoning, while enabling self-interpretable visualization for human-in-the-loop debugging. Our code and runtime logs are available at https://anonymous.4open.science/r/VLAgent.

## 1 Introduction

Compositional visual reasoning tasks often involve a sequence of heterogeneous visual reasoning subtasks, and demand for multiple independently trained vision models or vision-language models (VLMs) to perform different subtask-specific visual reasoning. Furthermore, different visual reasoning tasks tend to require different compositions of multiple vision models in order to generate correct visual reasoning output. Hence, learning to perform diverse compositional visual reasoning tasks poses significant challenges to advanced large vision-language models, including GPT-4o, GPT-5. In this paper, we present VLAgent, a vision-language agent system that explores the neuro-symbolic approach to automatically breakdown each compositional visual reasoning task from end-users into a sequence of task-specific neuro-symbolic instructions in two stages. We argue that a neurosymbolic approach to compositional visual reasoning could be viewed as an attractive complementary representational learning framework to the advanced large vision-language models. Furthermore, we argue that to ensure high performance and high accuracy in reasoning output, a robust integration of neuro-symbolic learning for compositional reasoning requires the following four critical functional components to work in concert: plan generation, plan debugging and repair, verifiable plan execution with stepwise auditing, and output verification. By self-validation and self-refinement of neuro-symbolic instructions generated for compositional reasoning, prior to generating the final output, will notably fortify the generalization performance of VLAgent for complex visual reasoning.

To date, research activities have been engaged towards complex visual reasoning along two interrelated threads. Neural Module Networks (NMN) Andreas et al. (2016b); Hu et al. (2018; 2017); Johnson et al. (2017); Andreas et al. (2016a) are pioneering in the model training category. This line of work aims to tackle the challenges of end-to-end visual reasoning models by decomposing complex reasoning tasks into modular compositional subroutines through supervised learning with large labeled training datasets. NMN development shows that neural modular networks can significantly improve the interpretability of visual reasoning. Inspired by the ideas of neural modular networks, recent approaches are centered on zero-shot learning, instead of training with supervised learning. Pre-trained LLMs (open source or close source) are utilized to generate structured programs for performing end to end compositional reasoning. For example, ProgPrompt Singh et al. (2023a) generates executable programs to help robots perform vision-related tasks. ViperGPT Surís et al. (2023) and VisProg Gupta & Kembhavi (2023) aim to solve visual question & answer (VQA) problems with zero-shot learning. ViperGPT formulates Python programs based on existing Python libraries and VisProg uses LLM to generate program template embedding calls to external modules (pretrained models or preconfigured Python functions , more detailed related work in Appendix F). However, existing visual reasoning methods suffer from a number of limitations. *First*, LLM generated programs often produce non-existent modules or logically flawed execution program steps. *Second*, existing methods lack of capability for checking the validity of LLM generated program w.r.t. both the feasibility of generating runnable code and the correctness of reasoning steps. As a result, existing methods tend to fail miserably when the LLM generated program is ill-formatted or logically incorrect due to unwanted hallucination. *Finally*, existing approaches often hard-wire a pre-defined external module for each reasoning step, making their performance bounded to the performance of the weakest external module(s) in the end-to-end reasoning process.

Motivated by the above observations, we present VLAgent, a vision-language agent system for efficient end-to-end visual reasoning with three novel characteristics. *First*, the VLAgent develops a two-stage neuro-symbolic visual reasoning agent framework. Stage-1 managed by a front-end engine will utilize few-shot and Chain of Thought (CoT) in-context learning to finetune a pretrained LLM to learn to create a stepwise visual reasoning plan in the form of a structured logic program for any visual reasoning request from end-users. Stage-2 managed by a backend engine will support the mapping of each planning script to executable code and perform runtime execution to generate final reasoning output. *Second*, the VLAgent optimizes the two-stage neurosymbolic framework by developing the VLAgent SS-Parser, geared to improve the quality and correctness of LLM-generated planning scripts prior to generating executable code for runtime execution. This SS-Parser will inspect and repair syntactic and semantic errors detected in the planning program script. *Next*, the VLAgent enhances the generalization performance of the VLAgent by incorporating an output verifier to validate and refine compositional visual reasoning steps during runtime execution.

| Video | | | | | |
|---|---|---|---|---|---|
| Question | Did the woman from the driving seat get back in the car after dancing? | How are the identities of the people shown? | What did he do at the end of the video? | What did the lady with green headband do before she released the catapult? | Why does the person wearing bracelet swipe the food away from the baby in the middle of the video? |
| GPT-5 | ✗ | ✗ | ◯ | ✗ | ✗ |
| GPT-5-Thinking | ◯ | ✗ | ◯ | ✗ | ✗ |
| LLaVA-Video-7B-Qwen2 | ◯ | ✗ | ✗ | ✗ | ✗ |
| LLaVA-NeXT-Video-7B | ◯ | ◯ | ◯ | ✗ | ✗ |
| LLaVA-NeXT-Video-34B | ◯ | ◯ | ✗ | ✗ | ✗ |
| InternVL-3.5-8B | ◯ | ✗ | ✗ | ✗ | ✗ |
| InternVL-3.5-38B | ◯ | ✗ | ✗ | ✗ | ✗ |
| VideoLLAMA3-7B | ◯ | ✗ | ◯ | ✗ | ✗ |
| VLAgent | ◯ | ◯ | ◯ | ◯ | ✗ |

Figure 1: Examples illustrating performance of eight VLMs on NeXT-QA compared with VLAgent.

**Figure 1** illustrates by examples the effectiveness of VLAgent compared with eight popular VLMs, including GPT-5 and GPT-5-Thinking on NeXT-QA benchmark Xiao et al. (2021). VLAgent correctly solves four out of five cases. Extensive experiments are performed on six representative visual reasoning benchmarks. The results demonstrate that VLAgent consistently outperforms existing zero-shot learning methods by 3%-40% in terms of accuracy on image QA benchmarks and surpasses the representative zero-shot methods on video QA benchmarks by a margin of 7%-19%.

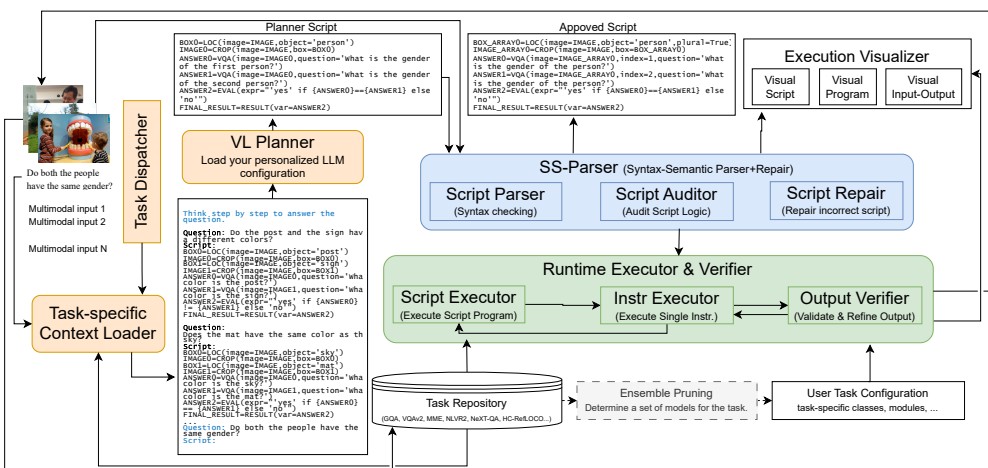

Figure 2: A two-stage Neurosymbolic architecture of the VLAgent system with front-end engine and backend engine working in concert.

## 2 METHODOLOGY

We give an architectural overview of the VLAgent design methodology in **Figure 2**. The front-end engine of VLAgent is shown on the left, consisting of task dispatcher, task-specific context loader, and visual reasoning planner (aka script generator). When VLAgent receives a visual QA query task (e.g., "*Do both the people have the same gender*"), the task-dispatcher will instruct the context-loader to compose the task-specific context with few-shot examples and CoT instructions as the prefix context for the visual reasoning planner to finetune a LLM with in-context learning (ICL). The LLM is either by default or chosen by the user of VLAgent in her configuration. Concretely, the planner will first prefix the visual QA query with the ICL context to compose a prompt query (see the middle rectangle under the planner by magnifying Figure 2) and then finetune the LLM via few-shot CoT in-context learning to generate a task-specific planning program script (an example in the top-middle rectangle). The planning script consists of a sequence of declarative program instructions, each is expressed as an assignment statement with output variable, per-instruction-specific visual reasoning module with module name and input parameters. All modules are either **external pre-trained models**, which can be by default or by user-choice in VLAgent configuration, or **internal modules** from the VLAgent Python library. **Figure 3** shows a list of core modules supported in the alpha version of VLAgent and used in the experiments reported in this paper. The first six rows are external pre-trained models marked in blue. The remaining modules marked in green, each corresponds to an internal module in the VLAgent Python library. `LOC`, `VQA`, `CAP`, `CROP` to `GET`, and `RESULT` are used for GQA, VQAv2, MME workloads.

The backend engine is the backbone of VLAgent. It takes both text input and visual input of each visual reasoning task, and examines the LLM-generated planning script through the SS-Parser (blue rectangle) and the Executor and Verifier (green rectangle). The **SS-Parser** (blue) consists of three main VLAgent modules: It first invokes the *Script Parser* and *Script Auditor* to examine the generated planning script and detect syntax errors and incorrect program logic. The *Script Repair* module is triggered to fix errors and generate the correct planning script. Once the SS-Parser completes the syntax-semantic checking and repairing, it will forward the correctness approved script to the next VLAgent backbone subsystem: the **Runtime Executor and Verifier**. It takes the SS-Parser-refined planning script with both text input and visual input and maps logic program script to the executable code, and then performs runtime execution with stepwise verification of reasoning correctness. It has three core modules. First, it invokes the *Script Executor* to generate executable codes to call external modules or internal library functions. Next, it runs each executable instruction by the *Instruction Executor*. By running the executable instructions corresponding to the sequence of declarative program steps contained in the planning script, the executor will summarize and generate the final answer. To ensure the robustness and generalization performance of compositional reasoning of VLAgent, we validate the output from the executor through the VLAgent output verifier, which performs inconsistency resolution with statistical confidence scores. The result with the highest score is returned as the final answer that is returned to the user. The task repository and execution visualizer are shared by both front-end and backend engines of VLAgent. The task repos-

itory stores not only the ICL examples and CoT instructions, but also the backend neurosymbolic modules and configuration for different types of visual reasoning tasks. For example, users can add a new task by registering new modules and corresponding ICL examples and CoT instructions. The execution visualizer utilizes human-interpretable visual API for debugging and human-in-the-loop feedback analysis in both stages of the neurosymbolic reasoning process, configurable in VLAgent through the configuration manager. With page limit, below we focus on the SS-Parser and the Plan Executor & Verifier to describe the VLAgent design methodology.

| Module Name | Pre-trained models/ Python Modules | Input | Output | Description |
|---|---|---|---|---|
| LOC | owlvit-large-patch14 owlv2-large-patch14 owlv2-large-patch14-ensemble grounding-dino-base TimeZero-Charades-7B (video) | Image/video, object name/event | A list of bounding boxes sorted in descending order of scores, or start and end frame of described event | Object detection given an object class, or event location given a description. |
| VQA | blip-vqa-capfilt-large vilt-b32-finetuned-vqa paligemma-3b-ft-vqav2-448 | Image, question | Answer | Answer visual attributes like color of objects in an image. |
| VIDQA | llava-video-7b-qwen2 VideoLLaMA3-7B InternVL3_5-8B | Video, question | Answer | Answer questions given a video. |
| CAP | Florence-2 | Image/video | Caption/Per-frame captions | Image captioning. When the input is a video, do per-frame captioning. |
| FIND | clip-vit-large-patch14 | Image. bounding box list, name | Bounding box | Select a bounding box whose content matches the name best. This allows to select a celebrity. |
| SELECT | gpt-4.1-mini | Question, answers from QA modules, choices | Choice index | Given a question and information collected from QA modules, select the best choice. Scores for each choice is also generated and recorded. |
| CROP | PIL.crop() | Image, bounding box list | Image | Crop at the first bounding box in the list. If the list is empty, return the original image. |
| CROP_LEFTOF | PIL.crop() | Image, bounding box list | Image | Get the image to the left of the first bounding box. If the list is empty, return the left part of the image. |
| CROP_RIGHTOF | PIL.crop() | Image, bounding box list | Image | Get the image to the right of the first bounding box. If the list is empty, return the right part of the image. |
| CROP_ABOVE | PIL.crop() | Image, bounding box list | Image | Get the image region above first bounding box. If the list is empty, return the top part of the image. |
| CROP_BELOW | PIL.crop() | Image, bounding box list | Image | Get the image region below first bounding box. If the list is empty, return the bottom part of the image. |
| EVAL | eval() | Expression | Result | Evaluate an expression written in Python syntax. |
| COUNT | len() | Bounding box list | Integer | Count the number of bounding boxes in the list. |
| GET | PIL.Image.size | Image | Bounding box list with one element | Get the bounding box (border) of an image. |
| VOTE | list() | Bounding box lists | Voted bounding box | Given lists of bounding boxes, select a bounding box with highest intersection. |
| CLIP | Video.clip() | Video, start and end of time an event | Clipped video | Clip a video. Here Video is a class in VLAgent to store a video. |
| LENGTH | cv2.VideoCapture.get() | Video | Duration of the video | Get the duration of the video in secs. |
| CLIP_BEFORE | Video.clip() | Video, start and end time of an event | Clipped video | Clip the video before the starting frame. |
| CLIP_AFTER | Video.clip() | Video, start and end time of an event | Clipped video | Clip the video after the starting frame. |
| CLIP_AROUND | Video.clip() | Video, time to clip around | Clipped video | Clip the video around the designated time. |
| RESULT | dict() | Variable name | Value of the variable | Return the value of a runtime variable as the result. |

Figure 3: Core Modules in VLAgent $\alpha$ release. NLVR2 uses `VQA`, `EVAL` and `RESULT`. VideoQA uses modules taking video input plus `SELECT` and `EVAL`. HC-RefLOCO (referring expression) uses `LOC`, `CAP`, `FIND`, `VOTE`.

## 2.1 SS PARSER

We describe the design of three core components of our SS-Parser in this section with illustrative examples. First, the *Script Parser* examines the LLM-generated planning script line by line to flag all syntax errors. The *Script Auditor* performs the semantic-level examination to detect semantically incorrect parameters and semantically incorrect logic sequence of the planning steps. The *Script Repair* is designed to fix the errors and generate a logically consistent sequence of instructions. Typical syntax errors are the cases where a non-existent module name or false input parameters were used in the function call to external modules or internal modules. A frequent logic failure that many LLMs suffer is to make up some input parameter(s) in the `LOC` module: such as locating an object of "standing" or "smiling", and alike. We provide auto-scanner and auto-fixer to spot error type and error location in the planning script, and correct those errors accordingly, including type-checking of all external and internal modules w.r.t. module names, module input parameter types and output variable names, plus some error types for domain-dependent visual reasoning tasks, e.g., video vs image. **Figure 4** illustrates by example the effect of SS-Parser on the overall performance of VLAgent. Consider the example user query "*Do both the people have the same gender*" with the visual image (the top left). The ground truth answer for this zero-shot VQA is given under the image in green color. The planning script generated by LLM (GPT-3.5 in this case) is given at the top with

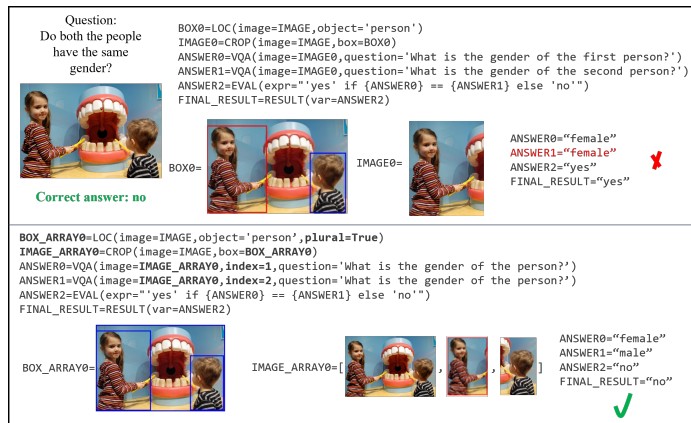

Figure 4: VLAgent SS-Parser corrects the reasoning error detected in the LLM-generated planning script.

six lines, calling `LOC`, `CROP`, `VQA`, `EVAL` and `RESULT` modules. The correct logic is to locate all the people in the image, crop them, ask the gender of each person and compare. Our SS-Parser spotted a logical error in the 4th line of the LLM-generated program, and fixed the error with the SS-parser refined script (shown in the bottom rectangle) prior to generating the executable. This results in correct final answer by VLAgent. In contrast, without using our SS-Parser, the execution of the LLM-generated script (in the top rectangle) resulted in a wrong answer. Concretely, by examining the script at the top, although the intention of the script looks correct, there is a critical logical error: IMAGE0 is obtained by `CROP` of only the bounding box with the highest confidence score, which is the girl in the cropped image. Hence, the girl's gender is asked twice. Given two people are asked for comparison in this visual reasoning query, our SS-Parser fixed this problem by explicitly using an array as the output variable for `LOC` instruction and considering the bounding boxes with the top two highest scores, followed by performing `CROP` for each of the two bounding boxes in the output array of `LOC` respectively, it will capture two people. Followed by using `VQA` to ask the gender question on each of the two cropped images in `IMAGE_ARRAY0`. This enables the `EVAL` module to compare the gender of the two people. The execution visualizer provides intuitive interpretation of the sequence of instructions performed for VLAgent to produce the correct final answer. Although we utilize this example to illustrate the SS-Parser's impact on VLAgent performance, the methodology of SS-Parser design, as we outlined earlier, involves a user-configurable set of pre-defined type-checking rules for both syntax and semantic errors w.r.t. external and internal module calls and the sequence of instruction steps with respect to the task-specific semantic context.

## 2.2 RUNTIME EXECUTOR & OUTPUT VERIFIER

Recall the green rectangle in the right middle of **Figure 2**, this VLAgent subsystem consists of three core components. The *Script Executor* takes as the input an SS-parser approved planning script and maps it into executable instructions line by line, preserving the sequence of planning steps. The *Instruction Executor* is called interatively to locate and run the corresponding external module or internal Python module. The output verifier performs stepwise evaluation of each visual reasoning instruction to determine whether and how ensemble methods are employed to verify and refine the per-instruction result, and caption-analysis is employed when the low confidence score is produced for an instruction-level visual-resoning subtask.

**Caption Verifier**. In image QA or video QA tasks, for each image or video frame being processed, if the result is in low confidence, we trigger the Caption-Verifier optimization to obtain the caption of the input image/video frame as a reference point for checking output consistency before generating our final answer. In the first prototype of VLAgent, we set Florence-2 Xiao et al. (2024) as the default external module to implement our caption verifier, reconfigurable in VLAgent initialization. The caption-verifier first generates a detailed caption of the image or the given video frame, which includes the key elements and their states. Next, it utilizes a pretrained LLM like GPT-3.5 (chat model) to assess whether the caption provides sufficient clues to infer the correct answer. If the caption contains the necessary information, the comparison of the caption-inferred answer is made with the script executor's output. If the comparison results in a match, indicating the consistency between the execution output and the caption-inferred answer, the final answer is regarded as highly reliable. Otherwise, VLAgent will analyze explicit details in the caption and the caption-derived answer is chosen as the final result if explicit and accurate clues are found in the caption analysis. Our ablation

study in Experiments section shows the improvement brought by the caption verification. We also provide a video QA example (Figure 7 of Appendix B) to show the detailed procedure of how our caption-verifier corrects a wrong result produced by our script executor.

**Ensemble Verifier**. VLAgent improves external-module reliability (e.g., LOC, VQA) by leveraging the fusion of multiple independently pre-trained models but develops ensemble selection algorithm to prune a pool of $N$ candidate models to a small subset of $M$ models for efficiency ($M << N$) Tekin et al. (2024). Let $P$ number of model calls to evaluate an ensemble, e.g., $P = 100$, each model $j$ returns $B_{ij}$ for LOC. We fuse all candidates per call to obtain $B_i$. The agreement and score are

$$I_{ij} = \frac{|Area(\cup B_{ij}) \cap Area(\cup B_i)|}{|Area(\cup B_{ij}) \cup Area(\cup B_i)|}, \qquad s_j = \frac{1}{P} \sum_{i=1}^{P} I_{ij}.$$

For VQA, set $I_{ij} = \mathbb{I}[A_{ij} \cap A_i \neq \emptyset]$; for VIDQA, set $I_{ij} = \mathbb{I}[choice_{ij} = choice_i]$. We cluster $\{s_j\}_{j=1}^{N}$ with K-Means MacQueen (1967), choose the number of clusters via the Silhouette criterion Rousseeuw (1987), and iteratively add the cluster(s) with the largest scores until reaching $M$ models for inference Chen et al. (2025). A detailed algorithm and notation is in Appendix E.2.

**Long Video Optimization.** Another critical optimization is resource-aware development of efficient neurosymbolic reasoning agent. Consider video QA tasks, we first develop video sampling methods through video partitioning into several chunks and then identifying important chunks via sampling. When using LOC module to locate an event in time dimension and use VIDQA module to perform video question answering, VLAgent feeds into these modules only a sequence of sampled frames rather than the entire video as input. For long videos, the sampling rate is usually much lower than the video frame rate. Therefore, we employ some optimizations to reduce or minimize the information loss from a low sampling rate. Suppose that we sample at most $f$ frames in these models, and we wish the sampling rate is at least $v$ fps. For a video of length $l$ (seconds), we uniformly divide it into $\lceil vl/f \rceil$ chunks. Note that $\lceil vl/f \rceil > 1$ in most scenarios under low sampling rate $f$, say 4 or 6 frames per video or per video chunk, we perform the following preprocessing: For each video, we sample $f$ frames uniformly and use a video QA model to generate the caption of the video. Then according to the question, an ensemble of multiple LLMs (e.g., GPT-4.1-mini) is used to judge whether only part of the video needs to be watched and leverages the sampled frame inference results to produce statistical scores on which chunks are more relevant to answering the question. We resample another $f$ frames from the video or from each chunk to perform the same procedure when the ensemble result is undesirable, e.g., no chunk is highly relevant or the entire video should be watched. We constrain this iterative procedure by at most $t$ iterations (e.g., $t = 3$). Given a small number of selected chunks needs to be watched for the visual reasoning question, we concatenate the selected chunks following their inherent temporal sequence and send it to the visual reasoning planner to follow the neurosymbolic procedure of VLAgent for compositional reasoning. Figure 7 in Appendix B illustrates this optimization with an example.

## 3 EXPERIMENTS

We report the evaluation of VLAgent and the comparison with the SOTA representative visual reasoning methods on six popular benchmarks: GQA Hudson & Manning (2019), NLVR2 Suhr et al. (2018), VQAv2 Goyal et al. (2017), MME Fu et al. (2024) (existence, position, color category), NeXT-QA Xiao et al. (2021) and HC-RefLOCO Wei et al. (2025). All experiments are conducted with H100/H200 GPU in a Python 3.9 environment.

### 3.1 EXPERIMENTAL COMPARISON RESULTS

**Table 1** compares VLAgent with representative image QA approaches in VLMs category and zero-shot methods on 4 popular ImageQA benchmarks. We observe that existing zero-shot methods, represented by ViperGPT Surís et al. (2023) and Visprog Gupta & Kembhavi (2023), exhibit a significant performance gap compared to supervised fine-tuning approaches on all 4 benchmarks. In comparison, VLAgent (zero-shot) achieves superior performance for NLVR2 and MME by $0.3 \sim 29.7\%$ and $1.0 \sim 30.3\%$ respectively, comparable performance for GQA and VQAv2. For VQAv2, VLAgent outperforms the three SOTA zero-shot methods (ViperGPT, VisProg and GENOME) and 10 VLM methods by $1.6 \sim 39.5\%$. Compare against the supervised model BLIP-VQA-Capfilt Li et al. (2022) (directly finetuned on VQAv2), VLAgent achieves 76.9% accuracy compared to the best VLM of 75.3%, reducing the gap to about 4%. For GQA, VLAgent achieves an accuracy of 61.9%, which outperforms all 3 zero-shot methods by $7.6 \sim 26.4\%$, and outperforms BLIP-VQA-Capfilt

(supervised) and 4 other VLMs by $3.8 \sim 17.4\%$. VLAgent offers on-par performance to InternVL3, and narrows the gap of all zero-shot methods to LLaVA-1.5-7B (70.3%), MiniCPM-V-4.5 (70.8%), Kimi-VL-A3B (71.1%) and Phi-3.5 (72.7%) by a large margin (61.9% vs $35.5 \sim 54.3\%$).

Table 1: Comparing VLAgent with 14 representative methods (VLMs and Zero-Shot) on 4 benchmarks. Improvement min and max are the lower and upper bound on improvement over 3 ZS and 10 VLMs methods.

| Method | NLVR2 | GQA | VQAv2 | MME |
|---|---|---|---|---|
| BLIP-VQA-Capfilt Li et al. (2022) (sup) | 44.8% | 54.5% | **81.2%** | 82.8% |
| Vilt-b32 Kim et al. (2021) (VLM) | 46.6% | 51.4% | 73.6% | 74.8% |
| PaliGemma2 Steiner et al. (2024) (VLM) | 44.4% | 44.5% | 46.5% | 61.6% |
| Gemma3-12B Sellergren et al. (2025) (VLM) | **73.8%** | 60.5% | 66.5% | 82.4% |
| SmolVLM Marafioti et al. (2025) (VLM) | 51.0% | 58.1% | 71.0% | 84.7% |
| InternVL3 Chen et al. (2024) (VLM) | 50.6% | 61.9% | 61.4% | 84.1% |
| Idefics2 Laurençon et al. (2024) (VLM) | 50.2% | 63.2% | 74.0% | 86.6% |
| LLaVA-1.5-7B Liu et al. (2023a) (VLM) | 50.3% | 70.3% | 71.0% | 85.0% |
| MiniCPM-V-4.5 Yao et al. (2025) (VLM) | 51.2% | 70.8% | 68.9% | 85.7% |
| Kimi-VL-A3B Team et al. (2025) (VLM) | 51.0% | 71.1% | **75.3%** | **87.4%** |
| Phi-3.5 Abdin et al. (2024) (VLM) | 64.5% | **72.7%** | 65.5% | 86.0% |
| ViperGPT Surís et al. (2023) (ZS) | – | 35.5% | 37.4% | 58.1% |
| VisProg Gupta & Kembhavi (2023) (ZS) | 69.3% | 54.3% | 72.3% | 85.3% |
| GENOME Chen et al. (2023) (ZS) | – | 44.7% | 60.5% | 77.9% |
| VLAgent (ZS) | **74.1%** | 61.9% | **76.9%** | **88.4%** |
| **Improvement on ZS (min)** | **+4.8%** | **+7.6%** | **+4.6%** | **+3.1%** |
| **Improvement on ZS (max)** | **+4.8%** | **+26.4%** | **+39.5%** | **+30.3%** |
| **Improvement (min)** | **+0.3%** | **-** | **+1.6%** | **+1.0%** |
| **Improvement (max)** | **+29.7%** | **+26.4%** | **+39.5%** | **+30.3%** |

We next evaluate the generalization capability of VLAgent on two recent video benchmarks: NeXT-QA Xiao et al. (2021) and HC-RefLOCO Wei et al. (2025). For NeXT-QA video understanding benchmark, we sample 200 questions per-type from the test set, resulting in a test set of 1493 questions. **Table 2** shows the comparison result of VLAgent with six VLMs and five VideoQA specific agent-based methods (zero-shot). VLAgent exhibits a high accuracy on all hard splits and achieves an overall accuracy of 76.0%, surpassing all 11 methods compared with the gain margin of $2.0 \sim 19.1\%$. For HC-RefLOCO Wei et al. (2025), representing the complex referring expres-

Table 2: Performance of VLAgent compared with 11 representative methods in VLMs category (top 6 rows) and zero-shot agent methods (middle 5 rows). Hard Split-T means temporal reasoning questions and Hard Split-C means causal reasoning questions. "Overall" represents for the overall accuracy of all types of questions.

| Method | Hard Split-T | Hard Split-C | Overall |
|---|---|---|---|
| LLaVA-Video-7B-Qwen2 Zhang et al. (2024b) | 60.4% | **69.5%** | 70.9% |
| LLaVA-NeXT-Video-7B Zhang et al. (2024a) | 54.8% | 65.3% | 65.8% |
| LlaVA-NeXT-Video-34B Zhang et al. (2024a) | 55.0% | 63.8% | 61.8% |
| InternVL-3.5-8B Wang et al. (2025) | 58.6% | 66.5% | 68.1% |
| InternVL-3.5-38B Wang et al. (2025) | 61.1% | 65.8% | 68.8% |
| VideoLLaMA3-7B Zhang et al. (2025) | **66.3%** | 68.0% | **72.2%** |
| ViperGPT Surís et al. (2023) (ZS) | 48.7% | 56.2% | 56.9% |
| SeViLA Yu et al. (2023) (ZS) | 59.6% | 58.5% | 64.0% |
| VideoAgent Wang et al. (2024) (ZS) | 60.0% | 68.3% | 66.1 % |
| TravelLER Shang et al. (2024) (ZS) | 56.9% | 65.4% | 66.0% |
| MoReVQA Min et al. (2024) (ZS) | 64.6% | 70.2% | 69.2% |
| VLAgent | **67.3%** | **74.8%** | **76.0%** |
| **Improvement on ZS (min)** | **+2.7%** | **+4.6%** | **+6.8%** |
| **Improvement on ZS (max)** | **+18.6%** | **+18.6%** | **+19.1%** |
| **Improvement (min)** | **+1.0%** | **+4.6%** | **+2.0%** |
| **Improvement (max)** | **+18.6%** | **+18.6%** | **+19.1%** |

sion benchmark on long descriptions, we sampled 1400 referring expressions from the test set to test VLAgent. Unlike traditional referring expression datasets like RefCOCO Kazemzadeh et al. (2014), which are already saturated in terms of performance due to the emergence of expression-based grounding models (e.g., OWLV2 Minderer et al. (2023) and Grounding Dino Liu et al. (2023b)

used in this paper), HC-RefLOCO grounds a person with fine-grained description of appearance, position, movement, and so forth, posing a greater challenge to grounding models. For each grounding task in HC-RefLOCO, we perform the caption-verification as follows: we first use the LLM to identify the person whose caption best matches the description; if no match is found, then we skip the verification, otherwise check whether the bounding box of this person substantially overlaps with the predicted result. If yes, the prediction is considered correct. Otherwise, we compare the caption of the predicted bounding box with the description to make the reasoning for the final alignment. We correct the bounding box only when it aligns significantly worse than the caption of the originally selected person. **Table 3** shows the performance of VLAgent on HC-RefLOCO compared to the 13 representative methods, of which SPHINX-v2-1K Lin et al. (2023) is the well-known SOTA method. Following the metrics in HC-RefLOCO, Acc0.5, Acc0.75, Acc0.9 and mAcc are used to measure and compare the performance. AccX means the ratio of test cases where the IoU between the predicted box and the ground truth is greater than X, and mAcc is the average value of Acc0.5 through Acc0.95 with a step size of 0.05. We observe that VLAgent surpass the 13 SOTA approaches compared in Acc0.75, Acc0.9 and mAcc, while using lightweight object detection models for inference time efficiency, and a performance gain over SPHINX-v2 (the best at Acc0.9) by about 7%.

Table 3: Comparison of VLAgent on HC-RefLOCO with 13 SOTA methods (zero-shot or VLMs).

| Model | Acc0.5 | Acc0.75 | Acc0.9 | mAcc |
|---|---|---|---|---|
| GPT-4V OpenAI (2023) | 17.4% | 2.6% | 0.3% | 5.5% |
| GroundingGPT Li et al. (2024) | 56.6% | 27.2% | 5.3% | 29.8% |
| Ferret 13B You et al. (2023) | 52.9% | 38.5% | 15.6% | 35.7% |
| KOSMOS-2 Peng et al. (2024) | 45.3% | 38.0% | 20.0% | 34.1% |
| Qwen-VL Bai et al. (2023) | 67.9% | 56.8% | 34.8% | 52.8% |
| OFA-Large Wang et al. (2022) | 70.5% | 61.6% | 44.0% | 58.1% |
| SPHINX Lin et al. (2023) | 77.5% | 61.0% | 27.0% | 55.4% |
| SPHINX-1K Lin et al. (2023) | 80.7% | 68.6% | 41.1% | 63.0% |
| SPHINX-v2-1K Lin et al. (2023) | **84.1%** | 77.1% | 56.2% | 71.7% |
| PixelLM 13B Ren et al. (2024) | 63.6% | 46.6% | 25.8% | 44.6% |
| LISA Lai et al. (2024) | 52.4% | 42.1% | 31.3% | 41.1% |
| PSALM Zhang et al. (2024c) | 61.7% | 53.6% | 40.2% | 51.1% |
| GlaMM Rasheed et al. (2024) | 66.1% | 56.9% | 44.2% | 55.0% |
| VLAgent | 82.6% | **77.4%** | **63.2%** | **73.9%** |
| **Improvement (min)** | - | **+0.3%** | **+7.0%** | **+2.2%** |
| **Improvement (max)** | **+65.2%** | **+74.8%** | **+62.9%** | **+68.4%** |

## 3.2 ABLATION STUDY

We compare the naive VLAgent (without SS-Parser+Caption-verifier+Ensemble verifier) with VLAgent+SS-Parser+Caption-verifier, and the full fledged VLAgent on four popular LLMs: gpt-3.5-turbo-instruct OpenAI (2023), Mistral-Small-24B-Base-2501 Jiang et al. (2023), GLM4-9B GLM et al. (2024), and Llama3-8B Grattafiori et al. (2024). **Table 4** reports the results on GQA. The performance of VLAgent improves progressively with the addition of SS-Parser and caption-verifier (row 2) and the addition of ensemble-verifier (row 3), compared to the naive version of VLAgent without SS-parser and output verifiers. The performance gains of VLAgent powered by our SS-Parser and Output Verifiers are consistent across the 4 popular LLMs for generating the planning scripts. Similar observations are made on other benchmarks as well (see Appendix D.1). An ablation study on inference latency is provided in Appendix D.2.

Table 4: Ablation study (GQA)

| Method | GPT 3.5 | Llama | Mistral | GLM |
|---|---|---|---|---|
| VLAgent naive | 54.4% | 54.1% | 55.2% | 54.7% |
| VLAgent +parser+cap-verf | 58.9% **+4.5%** | 56.9% **+2.8%** | 58.6% **+3.4%** | 58.2% **+3.5%** |
| VLAgent +parser+cap+ensemble | **61.9%** **+7.5%** | **58.7%** **+4.6%** | **60.7%** **+5.5%** | **60.4%** **+5.7%** |

## 3.3 PERFORMANCE COMPARISON BY VISUALIZATION

**Figure 5** illustrates the comparison results of **Table 1** with two examples per benchmark from GQA (columns 2 3), VQAv2 (columns 4 5) and MME (columns 6 7). Each example gives a non-trivial

| Image | | | | | | |
|---|---|---|---|---|---|---|
| Question | Do both the people have the same gender? | Does the stove top have the same color as the stove? | What is the name on the front of the train? | Are the girls sitting on a railing? | Is there a brown scarf in the image? Please answer yes or no. | Is the monitor on top of a person? Please answer yes or no. |
| Answer | no | no | trimet | no | no | yes |
| BLIP | ✗ (yes) | ✗ (yes) | ✗ (first) | ✗ (yes) | ✗ (yes) | ✗ (no) |
| VisProg | ✗ (yes) | ✗ (yes) | ✗ (first) | ✗ (yes) | ✗ (yes) | ✗ (no) |
| InternVL-3 | ○ (no) | ✗ (yes) | ○ (TriMet Max) | ✗ (yes) | ✗ (yes) | ✗ (no) |
| Phi-3.5 | ○ (no) | ✗ (yes) | ✗ (TRIOMET MAX) | ✗ (yes) | ○ (no) | ○ (yes) |
| GPT-4o | ✗ (unknown) | ✗ (yes) | ○ (TriMet Max) | ○ (no) | ○ (no) | ✗ (no) |
| GPT-5 | ✗ (unknown) | ✗ (yes) | ✗ (EXPO CENTER) | ○ (no) | ○ (no) | ✗ (no) |
| GPT-5-Thinking | ○ (no) | ○ (no) | ✗ (MAX) | ○ (no) | ○ (no) | ✗ (no) |
| VLAgent | ○ (no) | ○ (no) | ○ (TriMet Max) | ○ (no) | ○ (no) | ○ (yes) |

Figure 5: Visual comparison on six image QA examples

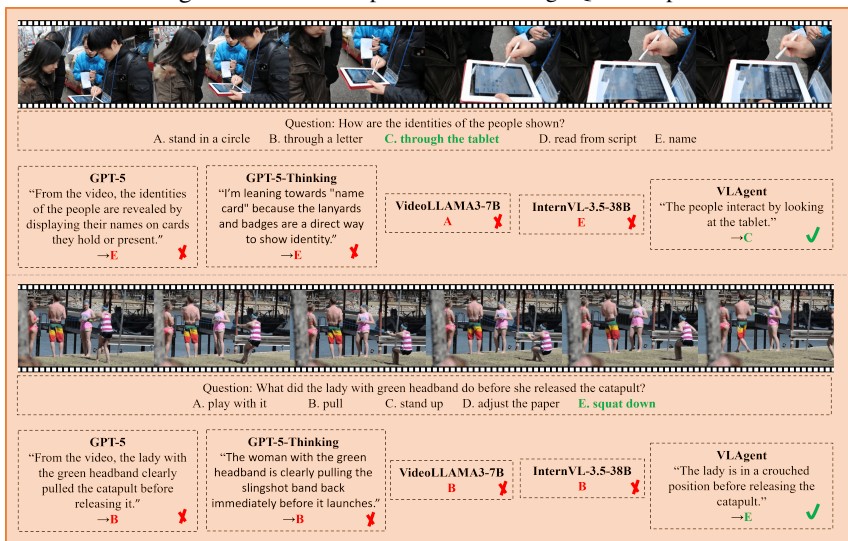

Figure 6: Two video QA examples comparing VLAgent with four selected approach in detail.

text-visual reasoning query. In all six cases, standard VQA models (incl. ensemble VQA method) fail to produce the correct answers, exposing their limitations in performing compositional visual reasoning tasks. Even with a massive amount of training data, GPT-4o failed on 3 out of the 6 queries. GPT-5 failed on 4 out of the 6 queries. Even GPT-5-Thinking only achieves good performance in 4 out of 6 cases. Consider the query in Column 7, the visual image shows clearly where the monitor is. However, GPT-5-Thinking failed on this visual reasoning. In comparison, VLAgent succeeds on all 6 cases by only utilizing lightweight pretrained models, empowered by its neuro-symbolic modularity design for robust compositional visual reasoning. **Figure 6** illustrate the effectiveness of VLAgent with two VideoQA examples in NeXT-QA. For each example, we provide the question, choices, answer, and output to compare VLAgent with 4 STOA methods. For Example-1 (top 2-rows), VLAgent succeeds by conducting compositional visual reasoning to make the correct choice. In comparison, the other four models (GPT-5. GPT-5-Thinking, VideoLLAMA3-7B, InternVL-3.5-38B) all failed to find visual clues to produce the correct inference results. For Example-2 (bottom 2-rows), VLAgent is able to make the correct choice of E (squat down). However, the other four models (GPT-5. GPT-5-Thinking, VideoLLAMA3-7B, InternVL-3.5-38B) all made the wrong choice B (pull) due to the failure to capture/identify the squatting body position.

## 4 CONCLUSION

We have presented VLAgent, a neurosymbolic approach to developing a visual-language agent system for compositional visual reasoning with three original contributions: (1) A novel two-stage neurosymbolic architectural design of VLAgent. (2) The use of SS-parser to empower VLAgent to detect and correct logic errors in the LLM-generated planning script. (3) The output evaluation via caption verifier and ensemble verifier to fortify the generalization performance of compositional reasoning. Extensive experiments conducted on 6 visual benchmarks show the effectivess of VLAgent in comparison with over 30 representative VLM/ZS methods.

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

APPENDIX CONTENTS

# A STATEMENTS

## A.1 REPRODUCIBILITY STATEMENT

We make every effort to ensure the results in this paper are reproducible.

- We provide a link of anonymous GitHub repository where the source code and runtime logs of VLAgent can be downloaded from.
- We provide details of modules used in VLAgent in Figure 3. In Appendix C, we provide the detailed settings of datasets and LLMs. In Appendix E, we provide full implementation details.
- Both in main paper and appendix, we provide figures as examples containing input, procedure and output to show how exactly VLAgent works.

## A.2 LARGE LANGUAGE MODELS (LLMS) USAGE STATEMENT

During the process of writing this paper, LLMs are used and only used for grammar checking and polishing of certain paragraphs.

# B ADDITIONAL EXAMPLES WITH VISUALIZATION

In this supplementary section, we provide additional examples with visualization to illustrate the main optimizations introduced by VLAgent in its backend engine, such as SS-Parser, Per-instruction output verifier via Caption Analysis and Ensemble based Visual Reasoning.

We first present an example video QA task, where the long video optimization and caption analysis plays the most important role in producing a correct answer. Then we use one example of image-based visual reasoning task to show how ensemble verifier contributes to the overall performance. In addition, we also include two failure cases as well as our analysis.

**Figure 7** illustrates a representative text-video reasoning task with the query "*Why did the girl start to shake her container horizontally after filling it up with water?*". The ground truth provided by NeXT-QA is given below the query with A as the correct answer (highlighted in green) out of the five multiple choice answers. In this example, the movement of the girl is fast, such that we can only infer it from the overall movement in the entire video. We use the four VLMs smaller than 8B in Table 2 to construct the model pool of the VIDQA module, which is used to answer questions based on a video. During ensemble pruning of the video QA task, we run VLAgent on 150 samples from the task dataset, and run the ensemble pruning process in Algorithm 2 to select three models. Finally, LLaVA-Video-7B-Qwen2 Zhang et al. (2024b), InternVL-3.5-8B Wang et al. (2025) and VideoLLaMA3-7B Zhang et al. (2025) are selected to construct the VIDQA module. Since the original video is too long, we divide the video into 5 chunks, with each chunk containing roughly 15 seconds, and use LLaVA-Video-7B-Qwen2 Zhang et al. (2024b) to get a detailed caption of each chunk. Then GPT-4.1-mini OpenAI (2025) is used to select chunks which may be relevent to the question. Chunks 2, 3 and 5 are selected, and they are concatenated together to form variable VIDEO. Meanwhile, our task planner generates a script to solve the problem. The script is then run with an initial variable VIDEO. In the figure, we divide the script execution process into two stages: keyframe locating and question answering. In the first stage, LOC is called to locate the timestamp where the girl has filled her container with water, and then clips the video after the starting frame. The clipped video is shown on the right of VIDEO0 in Figure 7, with keyframes annotated with red border. We can already see the girl is washing the container. In question answering stage, VIDQA is called to ask a bunch of questions on VIDEO0 and SELECT is called to choose the best answer. Unfortunately, video QA models do not provide enough information to distinguish A with C, and mistakenly votes C as the most possible answer. In verification step, we use per-frame caption model on VIDEO1, and answer A is clearly supported. Therefore, the answer is modified to answer A and VLAgent generates the correct final answer to this question.

**Figure 8** illustrates the effect of ensemble-verifier by an example from GQA with visualization. It shows that with ensemble verification, VLAgent can further improve the LOC performance using three external object detection models chosen by our ensemble-verifier. The cutting board bounding

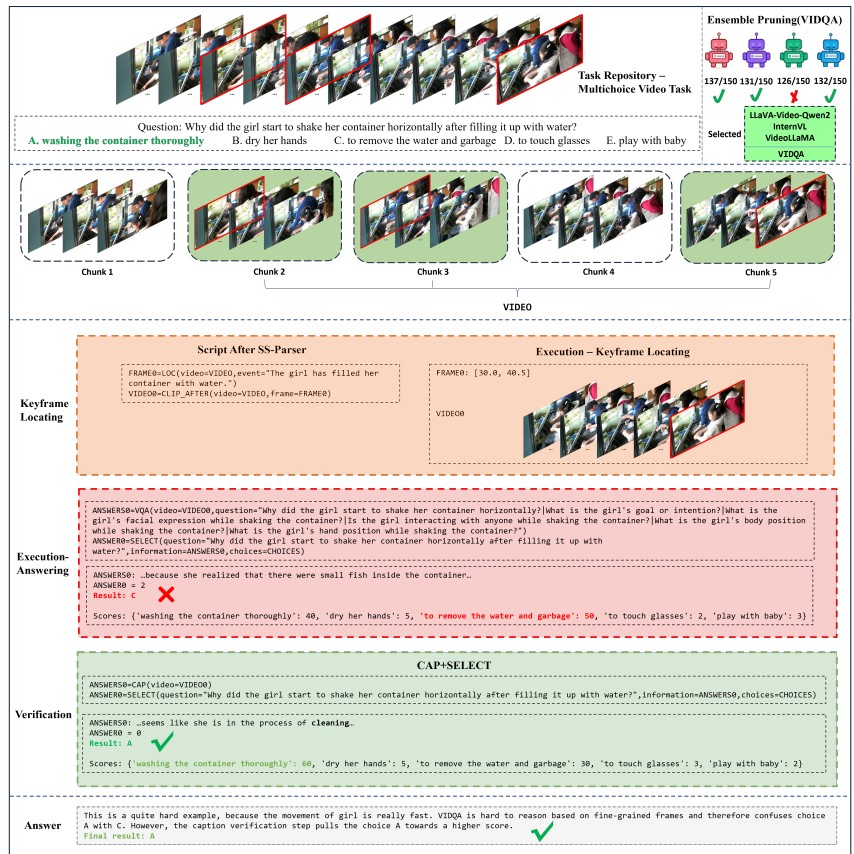

Figure 7: VLAgent example on video QA.

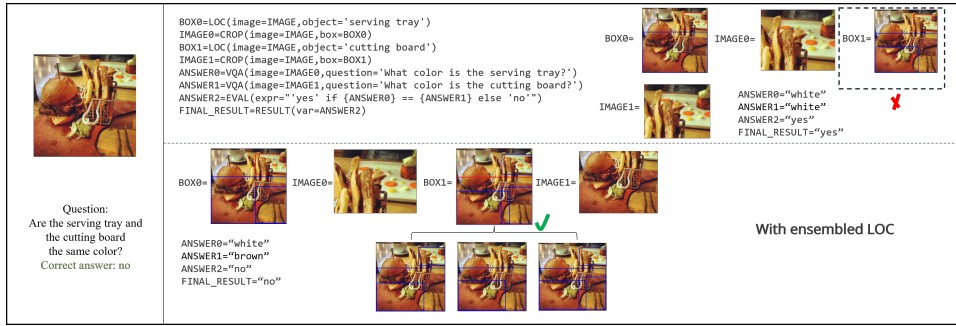

Figure 8: A GQA example illustrates the effectiveness of ensemble verification in VLAgent.

box in BOX1 gets the highest ensemble confidence score based on inconsistency resolution and fusion analysis. As we can see from the figure, the script locates the serving tray and cutting board, and then queries and compares their colors, which is correct. However, the default `LOC` module in VLAgent returns a wrong bounding box when it performs inference to locate the cutting board, making `IMAGE1` remains to be a serving tray, and thus outputs a wrong result. By leveraging two other `LOC` modules, VLAgent can verify the output of the original `LOC` module. As shown in the bottom of Figure 8, both of the additional `LOC` modules can successfully locate the cutting board. As a result, the cutting board is the top-1 bounding box ranked by the fusion confidence score instead of the serving tray, delivering the correct final result by VLAgent with ensemble boosting.

We also provide two failure scenarios from GQA in **Figure 9**, and **Figure 10**.

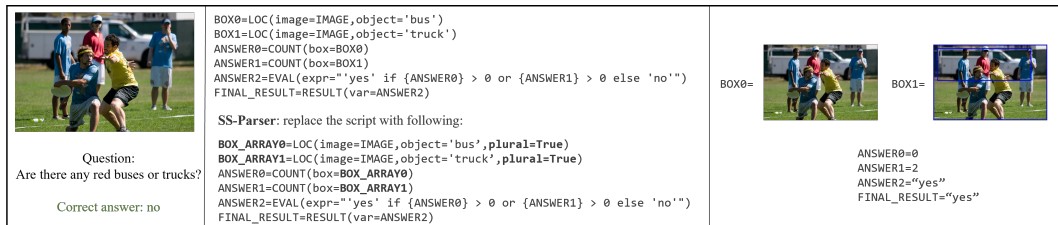

Figure 9: The script should examine the color of buses/trucks, but the SS-Parser fails to detect this type of error.

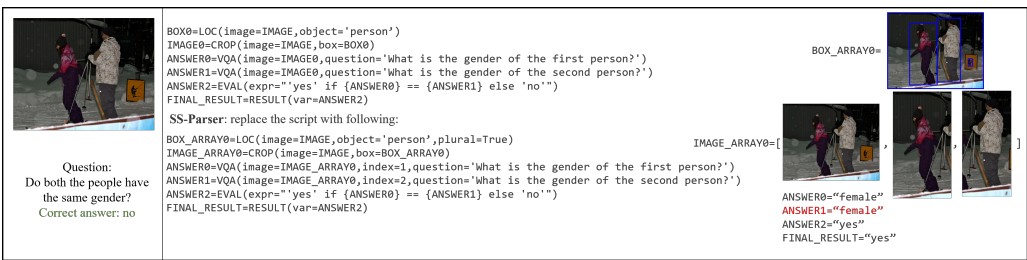

Figure 10: An example where VLAgent fails because the image is blurred.

In **Figure 9**, the script neglects to examine the color of the buses and trucks because the 20 in-context examples do not cover similar questions. Consequently, the SS-Parser fails to detect this type of logical error. This failure case motivates our future work in developing a more advanced semantic parser. For example, our revised SS-Parser can detect the error by judging the adjective word "red" in the question is not checked in the script.

Finally, in **Figure 10**, the VLAgent successfully corrects the errors in LLM-generated planning script via its SS-Parser and script-repair module. Unfortunately although the ensemble reasoning is leveraged, it fails when all of the three VQA models produce incorrect answers for the second person due to the fact that the image is too blurred and dark. This indicates that a more robust output verification method would improve the solution.

## C  DETAILED EXPERIMENTAL SETTINGS

In our experiments, we evaluate the performance of VLAgent and the baselines on six distinct datasets, each corresponding to a specific task. The datasets are described below:

- **GQA** Hudson & Manning (2019): A large-scale dataset for real-world visual reasoning and compositional question answering. GQA challenges models to understand complex scenes and answer questions that require multi-step reasoning. For example, a question might ask whether there is a boy to the left of a standing girl. For fair comparison, following the Visprog settings, we test VLAgent on a subset of the test_dev set by randomly selecting 20 questions per type, resulting in 1460 QA pairs.

- **NLVR2** Suhr et al. (2018): A dataset designed for natural language visual reasoning, consisting of paired images and textual statements. The task is to determine whether a statement accurately describes the visual content, thereby assessing both language understanding and visual grounding. We adopt the same setting as Visprog by evaluating on the entire balanced test set. After filtering out expired image links, 2316 samples remain.

- **VQAv2** Goyal et al. (2017): An improved version of the original VQA Antol et al. (2015) dataset, VQAv2 provides balanced question-answer pairs to mitigate language biases and foster deeper visual understanding. Widely used for benchmarking visual question answering models, we randomly sample 20 QA pairs per type from the validation set (the test set is not used as it lacks answer annotations). With 65 question types, this yields a total of 1300 QA pairs.

- **MME** Fu et al. (2024): A comprehensive benchmark designed to evaluate various aspects of multimodal reasoning, including math reasoning, code understanding, chart QA, existence, position, color, and more. In this work, we focus on a representative subset of the dataset—specifically the existence, position, and color categories—which require direct image-level reasoning. This selection aligns with our objective of evaluating visual reasoning capabilities independent of external components such as OCR or symbolic math engines. Categories heavily reliant on textual extraction or LLM-based arithmetic fall outside the scope of our image-centric evaluation. Our subset includes 580 QA pairs sampled from the official test set, providing a robust and targeted assessment of image-level reasoning performance.

- **NeXT-QA** Xiao et al. (2021): A large-scale *video* question answering benchmark targeting *temporal* and *causal* reasoning about human activities and events. Questions are multiple-choice and often require cross-frame understanding of before/after relations, intentions, and cause/effect. We evaluate on a subset of its test set, where we randomly sample 200 QAs per type, forming a subset of 1493 QAs.

- **HC-RefLOCO** Wei et al. (2025): A human-centric referring expression comprehension dataset with *long*, attribute-rich descriptions that combine appearance, pose, spatial relations, and actions. The task is to localize the target person (bounding box) that best matches the natural-language description, emphasizing fine-grained grounding over long sentences. We sample 1400 referring expressions from the official test split to form a subset to test VLAgent and other approaches.

Our experiments are conducted on a single H100 or H200 GPU in a Python 3.9 environment. We evaluate four LLM models for script generation:

- **GPT-3.5** OpenAI (2023): `gpt-3.5-turbo-instruct` API from OpenAI is employed to assess GPT's performance. For fair comparison, we change the LLM models of zeroshot baselines to GPT-3.5 as well if they are using older models.

- **Llama3-8B** Grattafiori et al. (2024): This model is loaded from HuggingFace at `meta-llama/Meta-Llama-3-8B`. Developed by Meta, it is a completion model with 8B parameters, a vocabulary size of 128K, and a context window of 8K - meeting the basic requirements for script generation.

- **GLM4-9B** GLM et al. (2024): This model is loaded from HuggingFace at `THUDM/glm-4-9b-hf`. It is a more advanced model, featuring 9B parameters, a vocabulary size of 152K, and a context window of 128K.

- **Mistral-Small-24B-Base-2501** Jiang et al. (2023): From HuggingFace at `mistralai/Mistral-Small-24B-Base-2501`, this model is loaded. Among the four LLMs, Mistral offers the best performance, with a parameter size of 24B (even larger than GPT-3.5, which has a reported parameter size of 20B Singh et al. (2023b)). Its vocabulary size is 131K, and its context window is 32K.

## D  ADDITIONAL ABLATION OF VLAGENT

### D.1  ABLATION OF TASK PLANNER LLM

**Table 5** reports the comparison results of full-fledged VLAgent with its naive version with only LLM-generated script and its runtime executor (w/o parsers and verifiers) with four popular LLMs as the LLM-script generator respectively and tested on all four ImageQA benchmarks. VLAgent consistently outperforms its naive version by a significant margin. In particular, for NLVR2 with GLM, the combo performs poorly with only an accuracy of 19.2% due to incorrect generation of LLM programs. In comparison, VLAgent achieves an accuracy of 58.6% with 39.4% gain margin. Similarly, for VQAv2, VLAgent significantly improves the accuracy with Llama by 14.3%, Mistral by 6.4%, GPT-3.5 by 4.6%, and GLM by 3.9%. For MME, the full-fledged VLAgent remains the top performer with 3.5% improvement on average. Given that the questions in the selected categories of MME are significantly simpler compared to those in GQA, the naive VLAgent can achieve very good accuracy. The improvement brought by the combo of SS-Parser and Output verifier is about $1.4 \sim 7.6\%$.

Table 5: Accuracy Comparison on 4 benchmarks (GQA, VQAv2, MME, NLVR2). Each benchmark is tested using four different LLMs as the corresponding initial task plan generators for both VLAgent naive (with only LLM script planner and executor) and VLAgent (the full fledged version with SS-Parser and caption and ensemble Verifiers). A total of 16 combos for VLAgent to compare with 16 combos of VLAgent naive, showing the consistent gain of SS-Parser and Output Verifiers.

| Agent Framework | Benchmark | GPT 3.5 | Llama | Mistral | GLM |
|---|---|---|---|---|---|
| VLAgent naive | GQA | 54.4% | 54.1% | 55.2% | 54.7% |
| VLAgent | GQA | **61.9%** | **58.7%** | **60.7%** | **60.4%** |
| Improvement | GQA | **+7.5%** | **+4.6%** | **+5.5%** | **+5.7%** |
| VLAgent naive | VQAv2 | 72.3% | 61.0% | 70.8% | 74.2% |
| VLAgent | VQAv2 | **76.9%** | **75.3%** | **77.2%** | **78.1%** |
| Improvement | VQAv2 | **+4.6%** | **+14.3%** | **+6.4%** | **+3.9%** |
| VLAgent naive | MME | 86.9% | 81.0% | 85.3% | 86.0% |
| VLAgent | MME | **88.4%** | **88.6%** | **88.8%** | **87.4%** |
| Improvement | MME | **+1.5%** | **+7.6%** | **+3.5%** | **+1.4%** |
| VLAgent naive | NLVR2 | 69.3% | 65.9% | 67.6% | 19.2% |
| VLAgent | NLVR2 | **73.2%** | **70.0%** | **74.1%** | **58.6%** |
| Improvement | NLVR2 | **+3.9%** | **+4.1%** | **+6.5%** | **+39.4%** |

## D.2 LATENCY TEST

The third experiment we want to report as a part of the ablation study is the latency of each core component and each optimization, especially the cost of ensemble verifier and caption verifier. **Table 6** reports the per-sample inference time of VLAgent on GQA, including its naive version (planner+executor only), and the individual core components within VLAgent. It is observed that empowered with all the SS-parser checking, repairing, and verifying mechanisms, VLAgent runs at 7.54 seconds per sample. Compared to the VLAgent naive which takes 3.24 seconds, the full-fledged VLAgent offers the inference latency at an acceptable range in practice. Also the breakdown of the components reveals that a majority of the added latency stems from the caption verifier, which invokes both an image captioning model and an LLM. In comparison, the ensemble fusion of vision models introduces only a modest overhead and the SS-Parser incurs negligible additional cost.

Table 6: Inference time per sample. VLAgent w. parallel means part of the caption verifier runs in parallel with the other VLAgent components.

| Component | Time Cost (s) |
|---|---|
| VLAgent Naive | 3.24 |
| VLAgent (all) Sequential | 7.54 |
| **VLAgent (all) Parallel** | **5.10** |
| Task Planning | 1.50 |
| LOC w/o ensemble verifier | 0.23 |
| LOC w/ ensemble verifier | 0.88 |
| VQA w/o ensemble verifier | 0.17 |
| VQA w/ ensemble verifier | 0.23 |
| SS-Parser | 0.00 (0.0016) |
| Caption Verifier | 4.01 |

Important to note is that the captioning and its analysis are independent of script generation and script execution, we can process them in parallel. For parallel implementation of caption-verifier, the total inference time of VLAgent can be reduced to 5.10 seconds per sample (a gain margin of 2.44 seconds). This also indicates that the checking and verifying mechanism only adds a tiny latency increase (less than 2s), while offering substantially improved visual compositional reasoning capability and improved interpretability.

# E   IMPLEMENTATION DETAIL

## E.1   SCRIPT PARSER & SCRIPT AUDITOR

Before passing the script to the executor, VLAgent checks for potential errors. The *script parser* looks for syntax errors; for example, it flags any script that uses a module name not supported by VLAgent. The *script auditor* checks for semantic-level errors. For instance, if the script attempts to locate "standing" in a cropped bird image to check whether a bird is standing, the auditor will note that the object name passed to the `LOC` module should be a noun or a phrase describing a noun. The auditor can detect such an error and return a corrected script.

**Table 7** summarizes the representative *detected conditions* and the *automatic repairs* applied by our two safeguards: the *script parser* and *script auditor*. For the script auditor, an expression like `== 'yes'` must be replaced with `== True` because we omit explicit type conversion in the template for simplicity. Instead, all variables are converted to their appropriate types inside the `EVAL` module: digit strings become numbers, "yes" and "no" become `True` or `False`, and other formats remain unchanged. However, the LLM is unaware of this and may still produce statements like `{ANSWER0} == 'yes'` to determine if a variable is "yes" or "no."

The script auditor may also add a `plural=True` flag in the `LOC` call if the object name is plural or corresponds to a plural word in the question. For example, if the question is "Are both people the same gender?" and `LOC` identifies a person, the auditor can detect that "person" corresponds to "people," meaning the bounding box should encompass a group of people. In such a case, the script auditor adds `plural=True` to the `LOC` call, and the bounding box of the entire image is returned if any person is detected. In implementation, the script parser and the script auditor can be implemented together, with a line-by-line check of the script, as shown in Algorithm 1.

Table 7: Representative issues and automatic repairs applied by the script parser and script auditor.

| Module | Detected Condition | Strategy |
|---|---|---|
| Script parser | Wrong script format
Non-existent module names
Non-existent variables
Syntax error in EVAL's expression | Replace with direct VQA call
Replace with direct VQA call
Replace with direct VQA call
Replace with direct VQA call |
| Script Auditor | LOC: Strange object names
LOC: plural object name
LOC: corresponding plural noun in question found
EVAL contains "== 'yes'"
EVAL contains "== 'no'" | Replace with direct VQA call
Add "plural=True" in LOC call
Add "plural=True" in LOC call
Replace it with "== True"
Replace it with "== False" |

On video QA task, SS-Parser adds the syntax check for `VIDQA`, `LENGTH`, `CLIP`, `CLIP_AROUND`, `CLIP_BEFORE`, `CLIP_AFTER`, `SELECT` modules, while for `LOC`, the input is changed to question along with a video. The script auditing part extracts the temporal keywords from the question and checks whether the usage of `CLIP` family models is correct. For referring expression, SS-Parser only does syntax check on supported modules.

## E.2   ENSEMBLE PRUNING

Consider `LOC` as an example. Suppose we have $N$ models to consider as candidate external modules. Instead of designing an ensemble of $N$ models, we consider only a small subset of $M$ models by ensemble pruning method Tekin et al. (2024) to avoid computation overhead.

We run our VLAgent on a small sample from the task dataset. Suppose we totally run `LOC` for $P$ times. Our ensemble pruning step consists of **three stages**: (i) Compute a score for each model to represent its overall performance; (ii) Use K-Means MacQueen (1967) to group models with similar performance together. Before clustering, we use Silhouette distance Rousseeuw (1987) to compute the best $K$. (iii) The models of the cluster with the highest scores are put to our candidate model list. If the number of candidate models is smaller than our desired $M$, go through (ii) and (iii) on remaining models.

---

**Algorithm 1:** High-Level Algorithm for SS-Parser. For simplicity, this algorithm only includes modules in GQA.

---

**Input:** program: multiline script;   question: user query;   module_list: allowed module names;   var_dict: runtime variable values (e.g. {`"IMAGE"`:None})
**Output:** approved_program: adjusted script or fallback

---

1 **begin**
2 | Split `program` into `lines`; Set `num_box_arrays`=0 and `num_image_arrays`=0;
  | **foreach** *(i, ℓ) in* `enumerate(lines)` **do**
3 | | Parse ℓ with `parse_step` into $(s, o, a)$; **if** *s not in* `module_list` **then**
4 | | | **return** fallback_script;
  | |
5 | | **switch** *s* **do**
6 | | | **case** *EVAL* **do**
7 | | | | Let *expr_fmt* = `a["expr"]`;
8 | | | | **if** *expr_fmt syntax error found in expr_fmt* **then**
9 | | | | | **return** fallback_script;
  | | | |
10 | | | | Set `var_dict[o]` = `eval(expr_fmt)`; In ℓ, replace `==` `'yes'` →`==` `True` and `==` `'no'` →`==` `False`;
11 | | | **case** *LOC* **do**
12 | | | | Set `var_dict[o]` = [[0,0,100,100]];
13 | | | | **if** `a["image"]` *not in* `var_dict` **then**
14 | | | | | **return** fallback_script;
  | | | |
15 | | | | Let *obj* = `eval(a["object"])`;
16 | | | | **if** *obj is not noun OR obj not mentioned in question* **then**
17 | | | | | **return** fallback_script;
  | | | |
18 | | | | **if** *obj is plural or corresponds to a plural noun in question* **then**
19 | | | | | `a["plural"]=True`;
20 | | | | | **if** `question` *contains any of* {`all, every, both, each`} **then**
21 | | | | | | $k \leftarrow$`num_box_arrays`**++**;
22 | | | | | | $o \leftarrow$"BOX_ARRAY_{k}";
23 | | | | | | For each CROP line using the bounding box: update bounding box name as new $o$; for each related VQA call: $m \leftarrow$`num_image_arrays`**++**, update image argument name to be `IMAGE_ARRAY_m`, add increasing index argument starting from 1;
24 | | | **case** *VQA* **do**
25 | | | | **if** `a["image"]` *not in* `var_dict` **then**
26 | | | | | **return** fallback_script;
  | | | |
27 | | | | Set `var_dict[o]` = '0';
28 | | | **case** *CROP/COUNT/RESULT/GET* **do**
29 | | | | Run *l*, updating `var_dict`; **if** *exception occurs* **then**
30 | | | | | **return** fallback_script;
  | | | |
31 | | | **else**
32 | | | | **return** fallback_script;
  | |
33 | | Replace line i of `lines` with the updated ℓ;
34 | Let approved_program = join(lines, "\n");
35 | **return** approved_program;

---

For each of the $P$ `LOC` instructions, we get the bounding box list of $M$ models, denoted as $B_{ij}$, where $1 \leq i \leq P$, $1 \leq j \leq M$. Let $B$ denote the final list of the $M$ bounding boxes produced by the ensemble fusion of $M$ external `LOC` models, , which serves as a *pseudo label*. Let $Area(B) := \{(x,y)|\exists b \in B, (x,y) \in b\}$ denote the union region of bounding boxes in bounding box list $B$. Hence, we can compute the confidence score $I_{ij}$ for each bounding box list $B_{ij}$ ($1 \leq i \leq P$,

$1 \leq j \leq M$), as follows:

$$I_{ij} = \frac{|Area(\cup B_{ij}) \cap Area(\cup B)|}{|Area(\cup B_{ij}) \cup Area(\cup B)|} \quad (1)$$

which is the Intersection of Union (IoU) between the $M$ bounding boxes in $B_{ij}$ and pseudo label. $|\cdot|$ is the area. We then use the average IoU of the $j$ th model as its score:

$$s_j = \frac{1}{P} \sum_{i=1}^{P} I_{ij} \quad (2)$$

After getting $\{s_j\}_{j=1}^{N}$, we go through stage (ii) and (iii) to iteratively select $M$ models to construct the model set for `LOC`. Refer to **Algorithm 2** for a detailed selection pseudo code. In Algorithm 2, line 2 to line 6 runs the VLAgent on $m$ samples from the task dataset. Line 7-10 goes through stage (i) to get a score of each model. Line 11-22 uses Silhouette distance Rousseeuw (1987) to select the best $K$, where $C(s_j)$ is the cluster $s_j$ belongs to, $a(j)$ is the intra-cluster distance for $s_j$, $b(j)$ is the inter-cluster distance for $s_j$, and $s(j)$ is the Silhouette distance of $s_j$. Line 23-26 runs $K*$-Means clustering, and adds the models in cluster $C^*$ which contains largest $s_j$s to the candidate model set $\mathcal{S}$, meanwhile removing them from the model pool $\mathcal{R}$.

---

**Algorithm 2:** Model Selection via IoU-based Scoring and K-Means Clustering

**Input:** $N$ models; $m$ samples from dataset; minimum number of models $M$

**Output:** $\mathcal{S}$: set of selected model indices

1   Initialize $\mathcal{S} \leftarrow \emptyset$, $\mathcal{R} \leftarrow \{1, 2, \ldots, N\}$
2   Sample $m$ data points and run `LOC` for $P$ total lines
3   **for** $i = 1$ **to** $P$ **do**
4      **for** $j \in \mathcal{R}$ **do**
5         Get bounding box list $B_{ij}$ from model $j$
6      Run ensemble algorithm to get pseudo label $B_i$
7   **for** $j \in \mathcal{R}$ **do**
8      **for** $i = 1$ **to** $n$ **do**
9         $I_{ij} = \frac{|Area(\cup B_{ij}) \cap Area(\cup B_i)|}{|Area(\cup B_{ij}) \cup Area(\cup B_i)|}$
10     $s_j = \frac{1}{P} \sum_{i=1}^{P} I_{ij}$
11   **while** $|\mathcal{S}| < M$ *and* $\mathcal{R} \neq \emptyset$ **do**
12     **for** $K = 1$ **to** $|\mathcal{R}|$ **do**
13       Run K-Means clustering on $\{s_j\}_{j \in \mathcal{R}}$ to get clusters $\{C_r\}_{r=1}^{K}$
14       **for** $j \in \mathcal{R}$ **do**
15         **if** $|C(s_j)| = 1$ **then**
16          $s(j) = 0$
17         **else**
18          $a(j) = \frac{1}{|C(s_j)|-1} \sum_{s_k \in C(s_j), k \neq j} |s_j - s_k|$
19          $b(j) = \min_{r \neq C(s_j)} \frac{1}{|C_r|} \sum_{s_k \in C_r} |s_j - s_k|$
20          $s(j) = \frac{b(j)-a(j)}{\max\{a(j),b(j)\}}$
21       $S(K) = \frac{1}{|\mathcal{R}|} \sum_{j \in \mathcal{R}} s(j)$
22     $K^* = \arg\max_{K \in \{1,2,\ldots,|\mathcal{R}|\}} S(K)$
23     Run $K^*$-Means clustering on $\{s_j\}_{j \in \mathcal{R}}$
24     $C^* = \arg\max_{r \in \{1,\ldots,K^*\}} \max_{j:C(s_j)=r} s_j$
25     $\mathcal{S} \leftarrow \mathcal{S} \cup \{j : C(s_j) = C^*\}$
26     $\mathcal{R} \leftarrow \mathcal{R} \setminus \{j : C(s_j) = C^*\}$
27   **return** $\mathcal{S}$

---

For other external modules like `VQA`, we run the same algorithm. The only difference is how $I_{ij}$ is computed. For `VQA`, $I_{ij} = \mathbb{I}[|A_{ij} \cap A_i| > 0]$, where $A_{ij}$ is the set of words generated by model $j$

on $i$-th data, and $A_i$ is the set of words in the ensembled answer. For VIDQA, $I_{ij} = \mathbb{I}[choice_{ij} = choice_i]$, where $choice_{ij}$ is the choice supported by model $j$ on $i$-th data, and $choice_i$ is the majority voted choice. The effectiveness of ensemble verifier is measured in the Ablation study in Section 3.2.

# F  RELATED WORKS

Our work is inspired by several pioneering projects in Neural Module Networks (NMNs) Andreas et al. (2016b); Hu et al. (2018; 2017); Johnson et al. (2017). NMNs were introduced to improve interpretability by decomposing visual reasoning into explicit sub-tasks. In NMN frameworks, a question is parsed into a layout of modular operations (e.g., find, filter, count), each of which is handled by specialized neural units, and the results are composed to produce the answer. This compositional design yields a step-by-step reasoning trace that is more transparent than monolithic end-to-end models. However, NMNs require supervised learning to train the module selection or layout predictor, often relying on ground-truth programs or strong annotations. Consequently, their generalization is constrained by the quality and quantity of training data, and hard to extend to new tasks without additional supervision. The recent progress in Neural Module Networks includes ViperGPT Surís et al. (2023), VisProg Gupta & Kembhavi (2023) and GENOME Chen et al. (2023). These recent projects formulate Python programs that invoke external trained models and Python library modules to obtain answers in visual reasoning tasks.

Our research is also inspired by recent research in LLM enhancement for visual reasoning Singh et al. (2023a); Surís et al. (2023); Yang et al. (2022); Gupta & Kembhavi (2023) without task-specific model training or supervised finetuning of fundation models Antonio Torralba (2024); Sharma et al. (2024); Huh et al. (2024). The visual inference methods prompt an LLM to output an explicit sequence of operations that can invoke pre-built executable modules. ProgPrompt Singh et al. (2023a) uses an LLM to produce robot task plans as executable code given high-level instructions. PICa Yang et al. (2022) introduces the representation of visual information as text via objects and their attributes detected, and it improves the in-context learning with the additional textual data to GPT-3 to obtain answer to a visual question. ViperGPT Surís et al. (2023) formulates visual questions as Python programs calling vision APIs to generate answers via code execution. VisProg Gupta & Kembhavi (2023) introduced a well-structured program instruction template for visual reasoning and an interpreter is to execute the external pre-trained vision moddel or a Python module in Python library. GENOME Chen et al. (2023) extends Visprog to unseen task scenarios by adding a process to create new modules and new in-context learning examples. However, most existing zero-shot approaches suffer from the problem of blindly entrusting LLM generated programs, instead of integrating error checking and repairing with result verification as the preconditions for invoking program execution, minimizing the detrimental effect of logical errors in LLM-generated programs, such as incorrect planning steps, non-existent external modules due to undesirable LLM hallucination.

