# OpenReview forum: "A Neurosymbolic Agent System for Compositional Visual Reasoning"
_ICLR.cc/2026/Conference — ICLR 2026 Conference Withdrawn Submission_

### Official Review · Reviewer_GmpR · 2025-10-22

**Soundness:** 3
**Presentation:** 1
**Contribution:** 3
**Rating:** 2
**Confidence:** 3

**Summary:**

The paper addresses compositional visual reasoning by proposing a two-stage framework (VLAgent) that integrates structured reasoning with executable code generation. In Stage 1, an LLM uses few-shot and chain-of-thought (CoT) in-context learning to generate a structured logic plan describing the reasoning process. In Stage 2, this plan is converted into executable code that is run to produce the final reasoning output. The method includes an SS-Parser module that inspects, audits, and repairs the generated reasoning scripts, as well as an execution verifier to validate and refine the final outputs. Together, these modules aim to improve correctness and reliability in multimodal reasoning tasks involving both images and videos.

**Strengths:**

- The authors propose a method for compositional visual reasoning that operates on images as well as videos, which shows improvements over existing methods. The method provides reasoning traces that improve the interpretability of the final output.
- The two-stage pipeline and the introduction of an SS-Parser and verifier to check and refine outputs are interesting and relevant contributions.
- The proposed ensemble pruning is an interesting approach for robustness when dealing with multiple models outputs

**Weaknesses:**

- My main concern is the presentation of the method. It is very hard to follow the different modules of the method and especially understanding the architecture, as the methodology mainly gives insights on what the modules are supposed to do without going into technical details how exactly it is done, at least on a formal level (what are inputs and outputs of the modules).
	- The front-end part is only very briefly explained. There are no additional details on the task dispatcher, task-specific context loader, and visual reasoning planner/script generator/VL planner. What prompts are used for it? Are there additional steps performed besides prompting? Where is the context coming from? These are critical details that should be included in the methodology and appendix.
	- Figure 2 of the method is visually dense and hard to read, abstractions could help improve the clarity here. Figure 3 is hard to read as well if you do not zoom in. A Latex table would probably be better readable. Generally, it might be better to put it in the appendix, while some examples from it could be explicitly mentioned and explained in the paper to give the reader a general idea of it.
	- Instead of explaining what the SS-Parser is doing in Fig 4, it would be more helpful to understand the technical details of how it is done there. From chapter 2, it is not clear how the parts of the SS-Parsers are implemented. The algorithm in the appendix helps, but it's hard to understand if this is just the script parser or also the auditor and repair included. How is the script repair implemented? The clarity here should be improved, and more necessary details should be added to the main paper (instead of just the appendix), at very least referencing the relevant parts in the appendix.

- The differences to previous visual programming systems (e.g., ViperGPT, VisProg) are not sufficiently discussed. A clearer articulation of novelty would strengthen the contribution.

- Some of the modules seem to have quite some computational overhead, especially when multiple models need to be called for the same task (model ensemble). This needs to be discussed in the paper, especially since some results are only marginally better than plain VLMs (Table 1).

**Questions:**

- What are the crucial parts in which your method is different from existing Visual Programming approaches like VisProg, ViperGPT?
- What are the default models for the modules of your method? Which are used in the experiments?
- When comparing to the baseline models, did you collect the results yourself, or did you take them from other papers or benchmarks? This should be clarified in the paper.
- You compare your results to "GPT-5" and "GPT-5-Thinking" in some of the qualitative examples. Which setup are you referring to by that? Does it mean you collected the results via the web app? Or if you used the API, which parameters for reasoning effort did you use?
- The notation in line 132 (“finetune LLM with in-context learning”) is misleading, this should be corrected, as in-context learning is not fine-tuning.
- Many of the paper’s examples focus on the “same gender” task, which may introduce unnecessary sensitivity. It would be more convincing to include examples with more relevant and less potentially sensitive queries.

---

### Official Review · Reviewer_Zjb7 · 2025-10-28

**Soundness:** 2
**Presentation:** 1
**Contribution:** 2
**Rating:** 2
**Confidence:** 4

**Summary:**

This paper introduces a specific pipeline for visual reasoning based on the idea of tool-calling and moderator "agent" modules. The authors compare their approach to several VLM and neuro-symbolic VLM approaches on altogether six different datasets.

**Strengths:**

The general idea of a neuro-symbolic agent approach is valuable for the community also the experimental evidence seems to show improvements over baselines.

**Weaknesses:**

However I find it very difficult to assess what exactly the core contribution or claim of this work is. The proposed VLAgent seems to incorporate so many different module that aren't properly described how they work overall, but also how they are implemented. This makes it quite hard to assess the significance of the results. E.g., if there are potentailly so many different models within VLAgent is the comparison to the other baselines fair? Perhaps the authors could specificy again what the core idea is of VLAgent and what the experiments show? I.e. what specific benefit do the experiments highlight?

The biggest issue for me is the missing or poorly described information on the method itself. E.g. what module is being trained, what is not? What are all of these modules based on? I still don't understand what these modules are, e.g., the task dispatcher? Even the section on the SS Parser (sec. 2.1) does not make it clear to me what the three submodules are and how they work, e.g. the Script Auditor?

In context of the previous remark: What is the underlying idea of this specific pipeline, i.e. what is the core idea of VLAgent?

The authors also only describe a few modules in detail as the argue due to space limits. I think this is tricky as missing this information makes it more tricky to understand the paper and its significance/contribution. I suggest compressing the text, as it is rather long at times, removing some of the long tables which don't seem to give much value or at least put the missing information in the appendix.

The figures are very difficult to read and follow. Most of them are in poor quality (low resolution) and contain far too small text.

Minor: the paper is essentially written as one wall of text, which is quite difficult to read and follow. I suggest adding more paragraphs.

**Questions:**

Fig1: I am not sure what the real benefit is of having figure 1 where it is without knowing much about VLAgent yet at that point.

In Fig 2: What does personalized LLM mean?

What is the benefit of Figure 3? What am I supposed to get out of this?

---

### Official Review · Reviewer_ZfUe · 2025-10-30

**Soundness:** 3
**Presentation:** 2
**Contribution:** 3
**Rating:** 6
**Confidence:** 2

**Summary:**

This paper introduces VLAgent, a two-stage neuro-symbolic vision–language agent for compositional visual reasoning. It combines LLM-based symbolic plan generation with executable neural–symbolic modules, featuring an SS-Parser for logic repair and an Output Verifier for stepwise validation. Experiments on six benchmarks show strong zero-shot gains and improved interpretability.

**Strengths:**

- The paper introduces a well-structured neuro-symbolic framework that clearly separates planning and execution stages. This design improves modularity and interpretability, allowing each stage to be analyzed, debugged, and extended independently.

- The proposed SS-Parser effectively detects and repairs syntax and logic errors in LLM-generated plans, addressing a common failure mode of LLM-based program generation,

- The model achieves strong zero-shot performance across multiple visual reasoning benchmarks.

- The approach provides interpretable, step-by-step reasoning that supports human-in-the-loop analysis. The visual scripts and execution traces make the reasoning process transparent, enabling diagnostic inspection and trustworthiness.

- The ablation study convincingly shows the contribution of each component, including the SS-Parser and Output Verifier. The incremental performance gains validate the necessity and effectiveness of these modules rather than attributing success solely to scale.

**Weaknesses:**

- The multi-stage architecture introduces additional computational and implementation complexity.
Each step of planning, parsing, repairing, and verifying adds latency and engineering cost, which may limit real-time or large-scale deployment. Additional clarification on the computational cost would enhance the paper's clarity.

- The evaluation is mainly limited to visual QA tasks, which restricts the demonstrated generality of the framework.
Broader reasoning domains (e.g., robotics, text–image synthesis) would strengthen claims of general neuro-symbolic capability.

- The approach heavily depends on large LLMs, and scalability to smaller or open-source models is not analyzed.
This dependency raises concerns about reproducibility, accessibility, and efficiency in resource-limited environments.

- The paper claims interpretability benefits but does not provide quantitative measures or user studies to support them.
While qualitative visualizations are shown, there is no formal metric or user evaluation to substantiate improved transparency or usability.

- The related work is currently placed in the appendix, but it should be included in the main text to better contextualize and position the proposed approach within existing research.

**Questions:**

- How do the authors quantitatively evaluate the claimed interpretability of VLAgent beyond qualitative visual examples?
- What is the computational overhead introduced by the proposed approach compared to direct LLM reasoning?
- How well does the proposed system generalize to non–visual QA domains or tasks beyond the evaluated benchmarks?

---

### Official Review · Reviewer_3Z8t · 2025-10-31

**Soundness:** 2
**Presentation:** 1
**Contribution:** 2
**Rating:** 2
**Confidence:** 3

**Summary:**

This paper introduces VLAgent, a neuro-symbolic framework designed for efficient compositional visual reasoning. The proposed system addresses three key limitations prevalent in existing methods:
1. The generation of programs containing non-existent modules or logically flawed execution steps.
2. Hallucination-induced issues, where outputs are ill-formatted or semantically incorrect.
3. Overall performance being constrained by the weakest external module in the pipeline.

To overcome these challenges, VLAgent integrates three novel components:
1. A visualization-enhanced two-stage framework that structures the reasoning process.
2. An SS-Parser which scrutinizes the syntactic and semantic correctness of generated planning scripts.
3. An execution verifier responsible for validating and iteratively refining compositional reasoning outcomes.

Extensive evaluations on six diverse visual benchmarks demonstrate VLAgent's effectiveness, where it consistently surpasses a dozen state-of-the-art models, establishing a new benchmark in compositional visual reasoning.

**Strengths:**

Detailed evaluation.
compares VLAgent with representative image QA approaches in VLMs category and zeroshot methods on 4 popular ImageQA benchmarks.
We next evaluate the generalization capability of VLAgent on two recent video benchmarks.
detailed ablation study.

The paper conducts a comprehensive evaluation of VLAgent across multiple dimensions.
First, VLAgent is compared against representative image QA approaches on four popular ImageQA benchmarks.
Subsequently, the model's generalization capability is evaluated on two video benchmarks.
Finally, a detailed ablation study is provided to quantify the contribution of each key component within the system.

**Weaknesses:**

### Clarity & Motivation
- The paper's organization hinders comprehension. The introduction primarily lists limitations and corresponding solutions without adequately explaining the underlying motivation—specifically, why the proposed two-stage framework and output verifier are effective in addressing the stated problems. This lack of conceptual linkage makes the argument feel incoherent.

- Similarly, the abstract enumerates contributions but fails to articulate their core motivation.

- Figure 2 contains excessive text and fails to visually emphasize the operation of the novel modules.

### Technical Explanation & Evidence
Several key technical claims lack sufficient explanation and empirical support.

- Two-Stage Framework: The authors should clarify the mechanism by which the two-stage framework specifically mitigates the generation of non-existent modules and logically flawed steps. Also, demonstrative examples or visualizations tracing the refinement process would substantiate this claim.

- Output Verifier: The role of the output verifier in alleviating the "weakest external module" bottleneck requires elaboration. Is the improvement achieved through a voting mechanism that automatically filters out inferior results? Visual evidence illustrating this filtering process and its impact on the final output would strengthen this point.

### Performance
A 5-second latency (Line 1077) appears long for practical applications.

**Questions:**

In Line 274, it is mentioned that the verifier prunes a pool of N candidate models down to a small subset M (with M << N) for efficiency, and that P denotes the number of model calls required to evaluate an ensemble.
What is the concrete values used for N, M, and P?

---

### Note · Authors · 2025-11-25

**Comment:**

We thank the anonymous reviewers for their time in reviewing this paper and will use their suggestions to improve this work in the future.

**Withdrawal Confirmation:**

I have read and agree with the venue's withdrawal policy on behalf of myself and my co-authors.